# Construction of a fiber-optically connected MEG hyperscanning system for recording brain activity during real-time communication

Hayato Watanabe[1,2,3☯], Atsushi Shimojo[3,4☯], Kazuyori Yagyu[1], Tsuyoshi Sonehara[5], Kazuyoshi Takano[6], Jared Boasen[3,7], Hideaki Shiraishi[4], Koichi Yokosawa[3]*, Takuya Saito[1]

1 Department of Child and Adolescent Psychiatry, Hokkaido University Hospital, Sapporo, Hokkaido, Japan, 2 Department of Child Studies, Toyooka Junior College, Toyooka, Hyogo, Japan, 3 Faculty of Health Sciences, Hokkaido University, Sapporo, Hokkaido, Japan, 4 Department of Pediatrics, Hokkaido University Graduate School of Medicine, Sapporo, Hokkaido, Japan, 5 Research and Development Group, Hitachi Ltd., Sapporo, Hokkaido, Japan, 6 Graduate school of Health Sciences, Hokkaido University, Sapporo, Hokkaido, Japan, 7 Tech3Lab, HEC Montréal, Montreal, Quebec, Canada

☯ These authors contributed equally to this work.
* yokosawa@med.hokudai.ac.jp

**Data Availability Statement:** The underlying data of our results are: Photo of digital oscilloscope monitor showing TTL signals (S1 Fig), Photos of

## Abstract

Communication is one of the most important abilities in human society, which makes clarification of brain functions that underlie communication of great importance to cognitive neuroscience. To investigate the rapidly changing cortical-level brain activity underlying communication, a hyperscanning system with both high temporal and spatial resolution is extremely desirable. The modality of magnetoencephalography (MEG) would be ideal, but MEG hyperscanning systems suitable for communication studies remain rare. Here, we report the establishment of an MEG hyperscanning system that is optimized for natural, real-time, face-to-face communication between two adults in sitting positions. Two MEG systems, which are installed 500m away from each other, were directly connected with fiber optic cables. The number of intermediate devices was minimized, enabling transmission of trigger and auditory signals with almost no delay (1.95–3.90 $\mu$s and 3 ms, respectively). Additionally, video signals were transmitted at the lowest latency ever reported (60–100 ms). We furthermore verified the function of an auditory delay line to synchronize the audio with the video signals. This system is thus optimized for natural face-to-face communication, and additionally, music-based communication which requires higher temporal accuracy is also possible via audio-only transmission. Owing to the high temporal and spatial resolution of MEG, our system offers a unique advantage over existing hyperscanning modalities of EEG, fNIRS, or fMRI. It provides novel neuroscientific methodology to investigate communication and other forms of social interaction, and could potentially aid in the development of novel medications or interventions for communication disorders.

digital oscilloscope monitor showing auditory signals (S2 Fig), mat file of visual signals (S1 File), and Electrophysiological data obtained by human subjects (S2 File). All files are available from the figshare database as follows: (S1 Fig) doi: 10.6084/m9.figshare.19127282 (S2 Fig) doi: 10.6084/m9.figshare.14872785 (S1 File) doi: 10.6084/m9.figshare.14872827 (S2 File) doi: 10.6084/m9.figshare.19127285.

**Funding:** This Research was supported by Strategic Research Program for Brain Sciences by Japan Agency for Medical Research and Development JP20dm0107567, The Watanabe foundation, and JSPS KAKENHI Grant Number 20H04496. The funders had no role in study design, data collection and analysis, decision to publish, or preparation of the manuscript.

**Competing interests:** The authors have declared that no competing interests exist. Employment of TS by the Research and Development Group, Hitachi Ltd. does not alter our adherence to PLOS ONE policies on sharing data and materials.

## Introduction

Real-time, face-to-face communication between two people is spontaneous and dynamic, and likely involves both cooperative and competitive brain responses. To properly capture these neural processes in both communicating parties requires a simultaneous and synchronous brain imaging technique known as hyperscanning. Hyperscanning studies regarding forms of human communication and social interaction have been widely reported in the neuroimaging modalities of functional magnetic resonance imaging (fMRI) [1–8], functional near infrared spectroscopy (fNIRS) [9–16], and electroencephalography (EEG) [17–23]. Among these, EEG has the highest time resolution, on the order of milliseconds, which is highly advantageous in hyperscanning studies regarding communication. However, despite advances in source localization techniques for EEG signals, the spatial resolution of EEG remains poor in comparison to other neuroimaging modalities. Alternatively, magnetoencephalography (MEG) has temporal resolution identical to EEG, and far superior spatial resolution due to the fact that magnetic fields are undistorted by cranial bone and tissue. Nevertheless, successful implementation of MEG hyperscanning for studying communication or social interactions between two adults remains comparatively rare [24–27].

The reason for the comparative rarity of MEG hyperscanning systems for communication studies likely stems from the fact that MEG devices themselves are rare, and thus rarely in close enough proximity to permit audiovisual signal transmission at latencies sufficiently low for natural communication. Baess et al. [24] avoided the issue of latency with their Network Time Protocol (NTP)-synchronized MEG hyperscanning system by foregoing video transmission, and only communicating audio via an Integrated Services Digital Network telephone landline. This method reportedly resulted in a local audio transmission delay of 4.7 ms, and a lab-to-lab audio transfer time of 12.7 ms at a distance of approximately five kilometers. In further adaptation of the same system, Zhdanov et al. [25], succeeded in transmitting both audio and video signals via a customized User Datagram Protocol at transmission latencies of $50 \pm 2$ ms and $130 \pm 12$ ms (mean $\pm$ standard deviation), respectively. With this level of latency, they report that nine pairs of adult subjects were able to synchronize right hand movements at an accuracy from 215 ms to as low as 77 ms. Meanwhile, Ahn et al. [26] used a similar NTP synchronization technique to hyperscan with two EEG/MEG systems separated by a distance of 100 km. Although they report successful implementation of a verbal interaction task between two adults, the task was not designed to be time critical, and the inherent transmission latencies of the hyperscanning system are not reported.

For smooth social interactions, the limit for audio and visual one-way transmission delays has been reported at 100 ms and 500 ms, respectively [28, 29]. Meanwhile, accurate perceptual integration of audio and visual speech stimuli reportedly begins to decline when audiovisual misalignment exceeds 80 ms at the group level, and can be less at the individual level [30]. In terms of perceived quality of stimuli, it has been reported that audiovisual misalignment exceeding 20 ms causes discomfort, particularly if the audio precedes the video [31, 32]. In musical contexts, where temporal accuracy is extremely important, perceived quality of sound even in non-professionals has been reported to deteriorate with as little latency as 10 ms [33]. Therefore, although the latencies reported by Zhdanov et al. [25] are certainly low enough for smooth social interaction and accurate audiovisual integration of speech stimuli, further reduction in audiovisual latencies and misalignment is still desirable.

In this study, we present a newly established MEG hyperscanning system that offers marked improvements in audiovisual latencies and misalignment over previously reported systems. The system comprises two MEG devices directly connected via fiber optic cables, with a minimum number of specially-selected low-latency intermediate devices, and an audio delay line

(ADL) which permits synchronization of the transmitted audio and video signals. Here, we describe the constitution of the MEG hyperscanning system, and methods and results of evaluation of its audiovisual latencies.

## Materials and methods

### Fiber optics and MEGs

Our MEG hyperscanning system was constructed by connecting two MEGs installed at Hokkaido University Medical and Dental Research Building (site A) and Hokkaido University Hospital (site B) using 473 m of fiber optic cables (Fig 1). Transistor-transistor logic (TTL) signals were used to verify transmission latency between the two MEG devices. The TTL signals were produced by a PC installed at site A, and transmitted to the MEG data acquisition systems (MEG Acqs) at both sites. The MEG hyperscanning system had an audio/visual (A/V) transmission system, which facilitated realistic, face-to-face communication between participants at the two sites. The video system was unified to 1080p/60p. Audio signals were synchronized with video signals using the ADL.

### TTL setup

We used TTL signals to match the timing of measurements between the two sites. TTL signals were transmitted from site A to site B as follows.

1. A PC at site A produced TTL signals.

2. These TTL signals were subjected to electrical-optical conversion using a digital signal bidirectional optical/electrical conversion module (DPDVD16–002-OPT(M), Nanaboshi Electric Mfg. Co., Ltd.), which is shown as Optic I/O module A in Fig 1.

3. The converted optical TTL signals were transmitted via fiber optics to site B.

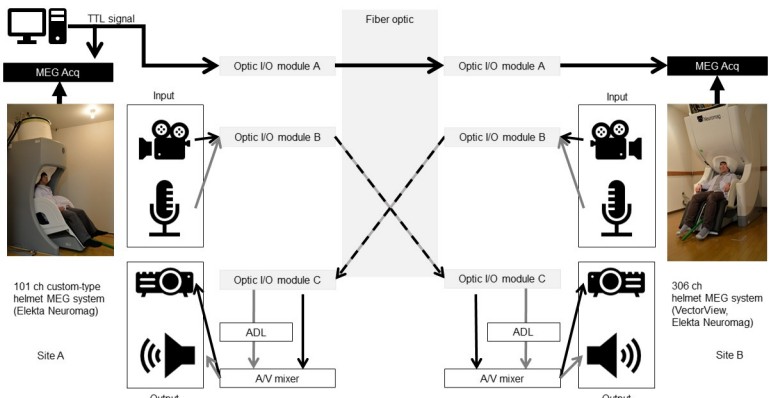

**Fig 1. Connection overview.** All signals were subjected to electrical/optic conversion, optical transmission, and optic/electric conversion using Optic Input-Output modules (Optic I/O module A–C). The TTL signal was output from the PC at site A and recorded by the MEG data acquisition systems of each site. The timing standards were set using the TTL signal. The audio/video signal input/output unit serves as the communication device. The video signal is optically transmitted from the camera and presented from the projector via an A/V mixer. The audio signal is transmitted through the microphone, its latency matched to that of the video signal using an audio delay line, and then presented from the speaker via the A/V mixer. Photos: with permission by the models.

4. The transmitted signals were decoded into electrical TTL signals using an identical conversion module at site B.

5. The decoded electrical TTL signals were received by the MEG Acqs at site B.

## Video setup

To ensure the potential to visualize small changes in the facial expressions of future participants, the video systems needed high resolutions and frame rates. Here, progressive scanning has advantages for motion recording and playing. Therefore, video signal transmission was unified to 1080p/60p. Video signals are transmitted from one site to the other site as follows:

1. At one site, video signals are sampled by an HD camera (GP-KH232A, Panasonic) in a shielded room, transmitted to the HD camera control unit outside the shielded room via a 15 m cable, and converted into HDMI signals.

2. The converted HDMI video signals are transmitted to a distributor (CRO-HD13, Imagenics).

3. The video signals are then converted to optical signals using an HDMI/DVI optical extender (Transmitter; CRO-FD24 TX, Imagenics), which is shown as Optic I/O module B in Fig 1.

4. The converted signals are transmitted via fiber optics to the other site.

5. Upon arrival, the signals are decoded to HDMI signals using an HDMI/DVI extender (Receiver; CRO-FD24 RX, Imagenics), which is shown as Optic I/O module C in Fig 1.

6. The decoded video signals are then transmitted to an A/V mixer (VR-4HD, Roland).

7. Video signals are then visually rendered by a projector (VPL-CH355, Sony) via the A/V mixer.

## Audio setup (with ADL synchronizaton)

As video signals are output in units of frames, the latency of video signals strongly depends on the presentation time of each frame. In contrast, audio signals are transmitted without frames; as a result, audio signals were expected to be transmitted more rapidly than video signals (See Results in detail). Therefore, to adjust the latencies of the audio signals, we additionally tested their output via ADL. Audio signals are transmitted from one site to the other site using ADL as follows:

1. At one site, audio signals are sampled using a monaural microphone (AT9904, Audio-Technica).

2. The sampled audio signals are transmitted to the distributor (DA-144, Imagenics).

3. The signals are embedded into HDMI signals and converted to optical signals using the HDMI/DVI optical extender together with video signals (Transmitter; CRO-FD24 TX, Imagenics).

4. The converted audio signals are transmitted via fiber optics to the other site.

5. The transmitted signals are decoded into analogue audio signals using the HDMI/DVI optical extender (Receiver; CRO-FD24 RX, Imagenics).

6. The decoded signals are transmitted to the A/V mixer via ADL (ADL-40, Imagenics).

7. The audio signals are played on a non-magnetic speaker (Audio Element N-20 in SSHP60X20, Panphonics) via the A/V mixer.

## TTL latency measurement

The standard signaling latency between the two sites was defined by a TTL signal. The latency of the TTL signal, which consists of durations of conversion and transmission (Fig 1), was measured as follows. A TTL signal generated by a PC at site A was recorded by a digital oscilloscope (Advantest, R9211E digital spectrum analyzer) at site A after a round trip to site B (loop back condition). The same TTL signal was directly recorded by the same digital oscilloscope without the round trip (direct condition). The time difference between those two conditions was evaluated. Half of the time difference was defined as the TTL signal latency. The sampling frequency of the digital oscilloscope was set 256.41 kHz (3.90 $\mu$s). The TTL signals were transmitted and recorded 100 times to confirm the reproducibility of our latency measurement.

## Video and audio latency measurement overview

The latency of video or audio signals caused by conversion, transmission, and passage of the signal through all intermediate devices. To measure the latencies of the video or audio signal from one site to the other, references were required to know the onset time of the signal. The reference signals also had inherent latencies. Therefore, the latencies of the reference signals were also evaluated.

Latencies were measured 100 times with a digital oscilloscope and averaged. Jitter was observable as distortion in the averaged latency. When evaluating jitter, latencies derived via the MEG Acqs at a sampling rate of 1,000 Hz were analyzed to determine their mode, average, variance, and range.

## Video latency measurement

Flashing LED lights and photodiodes to detect them were used to measure the latency of the video signal.

**Reference signal.**   The reference signal for video was a square signal which was generated by the output of the photodiode detecting the LED light. The square signal was directly transmitted from one site to the other site via the same pathway as the TTL signal described above, and input into the MEG Acqs at the receiver site. The sampling rate was 1,000 Hz.

**Measurement signal.**   The LED light was flashed 200 times over five sessions (total 1,000 times) at site A. The light was captured by the video camera at site A, and transmitted via all intermediate devices to site B, where the light was projected into the shielded room and detected by a photo diode. A square signal was generated by the output of the photodiode and input into the MEG Acqs at site B. This measurement process was performed in the opposite direction as well.

## Audio latency measurement and adjustment

Sine waves (250 Hz, 100 ms, 5 ms rise/fall) generated by a PC were used to measure the latency of the audio signal.

**Reference signal.**   The reference signal for audio transmission was a sine wave generated by a PC at site A. It was recorded by a digital oscilloscope at site A after a round trip to site B via optical analogue link (Transmitter, PE-1800TAF, Optex; Receiver, PE-1800RAF, Optex).

The loop-backed sine wave was compared with the original one on the same digital oscilloscope at site A.

**Measurement signal.** The sine wave signal generated by the PC at site A was split. One part was transmitted directly to site B and recorded on a digital oscilloscope. The other part was played on a non-magnetic speaker and sampled by a monaural microphone in the shielded room at site A. The signal captured by this microphone was then transmitted to site B where it, underwent digital audio conversion and passed through the A/V mixer. It was then re-played on a non-magnetic speaker and sampled by a monaural microphone in the shielded room at site B. The sampled signal was recorded on the same digital oscilloscope at site B. The audio waves of the two split signals were compared. This measurement process was performed in the opposite direction as well.

**Audio latency adjustment.** After determining the latencies of the audio and video signals, the latency of the audio signal was adjusted to the latency of the video signal by ADL, as appropriate. The minimum adjustment width of ADL was 1 ms.

### Electrophysiological experiment

One pair of subjects (23 year-old female and 25 year-old male) participated. Signed informed consent was obtained from both subjects before the experiment. The MEG recordings were approved by the Ethics Review Board of the Graduate School of Medicine at Hokkaido University.

The two subjects faced each other via the A/V devices and spoke words in turns according to timed cues. The speech audio signals from each site were transmitted to the opposite site with a 90 ms delay using the ADL to align them with the visual signal delay. MEGs were recorded during 128 speech exchanges of this alternate speaking protocol. The amplitude modulations of the alpha-band rhythms across all 128 exchanges were averaged and then normalized in each subject based on their average alpha amplitude over the period from -2,000 ms to -1,000 ms prior to the speech onset of the other subject (S2 File). The resulting normalized mean alpha activity was then mapped onto template brains. Data analysis was performed with Brainstorm [34].

## Results

### TTL latency

The time difference between the loop back and direct conditions were recorded by the digital oscilloscope as 7.80 $\mu$s for all signals (S1 Fig). Given that the sampling rate of the digital oscilloscope was 3.90 $\mu$s, this means that the signal latency of the loop back condition was later than 3.90 $\mu$s and shorter than 7.80 $\mu$s. Therefore, the latency of the direct condition, which was considered to be half the latency of the loop back condition, was evaluated to be 1.95–3.90 $\mu$s. No jitter was observed within this time resolution. The theoretical latency of the TTL signals was 2.88 $\mu$s, which is the sum of the integral of the speed of light over the transmission distance of 472 m (1.58 $\mu$s) and the time required for conversion (1.30 $\mu$s) by optical I/O module A (Fig 1). Thus, our measured latency coincides the theoretical one, and is much smaller than the highest temporal resolution of the MEG Acqs (1 ms at 1,000 Hz sampling).

### Video latency

**Reference signal.** Our evaluations revealed that it took 11.61 $\mu$s to generate the square signal from the output of the photodiode. Therefore, the latency of the reference signal was the

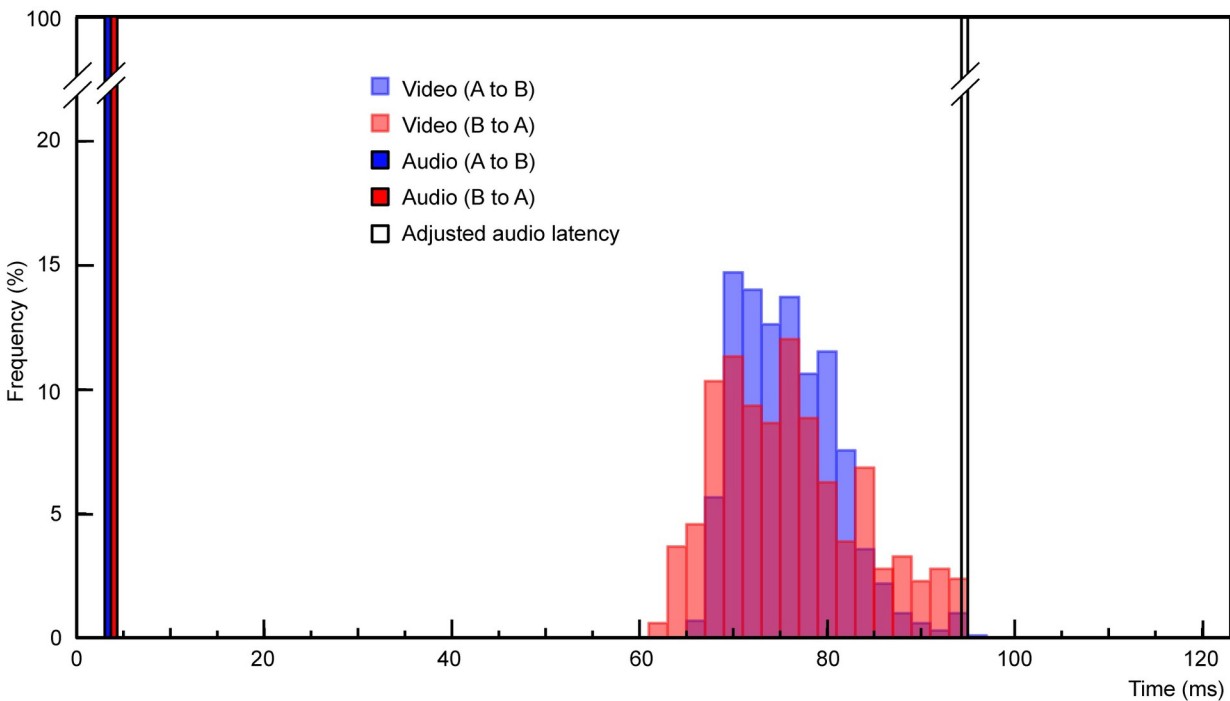

**Fig 2. Video and audio signal latency distribution.** Video and audio signal latencies from site A to site B and those from site B to site A are superimposed. Audio signal latencies (red/blue bars on the left) are short and have no jitter, while video signal latencies (red/blue bars on the right) are longer and have some jitter (mean: 76.85 ms, SD: 6.57 ms) ranging from 60–100 ms. To optimize the setup for natural communication, audio signals can be delayed to match the latency of the video signals (white bar).

sum of this delay of 11.61 $\mu$s and the direct latency of the TTL signal (1.95–3.90 $\mu$s). Effectively, the latency of the reference signal was negligibly short compared to the measurement signal.

**Measurement signal.** The latencies of the 1,000 LED light flash at both sites are summarized on a histogram with 2-ms bins (Fig 2, site A blue bars, site B red bars, S1 File). From site A to site B, the mode was 70–72 ms (mean = 76.76 ms, SD = 5.34 ms, and range = 66.42–97.42 ms); from site B to site A, the mode was 76–78 ms (mean = 76.94 ms, SD = 7.61 ms, and range = 63.42–95.42 ms). Here, the transmission takes 2.36 $\mu$s, which was calculated by the transmission speed of the HDMI cable 0.5 $\mu$s/100 m and cable length of 472 m, and conversion takes 400 $\mu$s in total (200 $\mu$s for optic I/O module B and C in Fig 1). As both sites had the same devices and set-ups, the latency distributions were nearly identical, ranging from 60 ms to 100 ms. These latencies are sufficiently short for natural communication, thereby meeting the objective of this system.

The processing latency of one of the intermediate devices, the A/V mixer, is 16.67 ms/ frame. This latency is small compared to the mean overall latency of about 77 ms. Hence, the majority of the video latency is implicitly caused by the camera and the projector. The signal transmission of the camera and the projector are 1080p/60p, i.e., one frame equates to 16.67 ms. Jitter was presumed to be caused by the frame of both the camera and the projector, and was therefore calculated as 33.34 ms (16.67 ms × 2 devices). The latency range of our measurement results (about 31.5 ms) closely coincides with this value.

## Audio latency and synchronization with video

**Reference signal.** The loop-backed sine wave was compared with the original one on the digital oscilloscope at site A. The latency, calculated as the half of the difference between the

two waves was 202.4 $\mu$s. This latency was negligibly short compared to the measurement signal. There was no distortion of the sine wave, based on visual inspection, thus indicating an absence of jitter.

**Measurement signal.**    A comparison of the audio waves of the two split signals demonstrated a constant latency of 3.13 ms (from Site A to Site B) and 2.78 ms (from Site B to Site A) with no jitter (Fig 2, red and blue bar, S2 Fig). The reason for the slight directional difference is not clear, but we suspect that it might depend on the distance between the microphone and the speaker at each site. Regardless, the minute directional difference (0.4 ms) is arguably not physiologically discernible, and the approximately 3 ms jitter-free latency in both directions is sufficiently low for natural communication.

**Audio latency adjustment.**    Audio signals from one site arrive at the other site about 74 ms earlier than the video signal. As mentioned previously, this situation is known to cause discomfort when viewing video [31, 32]. To correct this, and ensure that our system can be comfortably used for real-time audiovisual communication, ADL was used to increase the latencies of the audio signals to make them arrive just after the video signals. The ADL was set such that the audio signal latencies were increased by 90 ms, which is approximately two standard deviations above the mean video signal latency (76.85 ± 6.57 ms). Consequently, the ADL-adjusted audio signals had a mean latency of 93 ms (Fig 2, white bar).

## Electrophysiological experiment

Fig 3 shows normalized mean alpha-band amplitude modulation across all 128 speech exchanges for each subject averaged across the entire cortical surface (Upper), and that across both subjects and mapped onto the template brain (Lower). The brain activity of the subjects at both sites reflects that which is associated with listening, with time point 0 ms being the moment of speech onset of the opposite party. Alpha-band desynchronization was exhibited in both the site A and site B subject during listening. Notably, the desynchronization appears to have commenced before the speech onset of the opposite party, a sign that subjects could visually predict the onset of the opposite party's speech. The suppression was primarily concentrated in occipital and left temporal regions, indicating functional involvement of both the visual and auditory systems, and suggesting that each subject could visually predict the onset of the opposite party's speech.

## Discussion

We established an MEG hyperscanning system with an audiovisual interface capable of permitting real-time, face-to-face communication between two adults, and verified its TTL signaling and audiovisual transmission latency. The latency of the TTL signal (trigger) was orders of magnitude lower than the maximum temporal resolution of our MEG devices, essentially demonstrating simultaneous and synchronous recording onset for both MEG devices. Site-to-site audio signal latency was about 3 ms, in either direction, which is on par with the speed of transmission of telephone landline audio signals [25]. Moreover, audio latency was completely jitter free, and well below reported thresholds for human detection of musical quality deterioration, indicating that our system would additionally be suitable for communication paradigms based on musical stimuli [33]. Finally, the video signals had short latencies (60–100 ms) and small jitter (SD: 6.57 ms). We also conducted an electrophysiological study and confirmed that this hyperscanning system can reliably transmit A/V information and measure physiological signals.

The latencies and jitter values recorded here are the smallest ever reported for an MEG hyperscanning system. The additional verification of audio synchronization to video signals

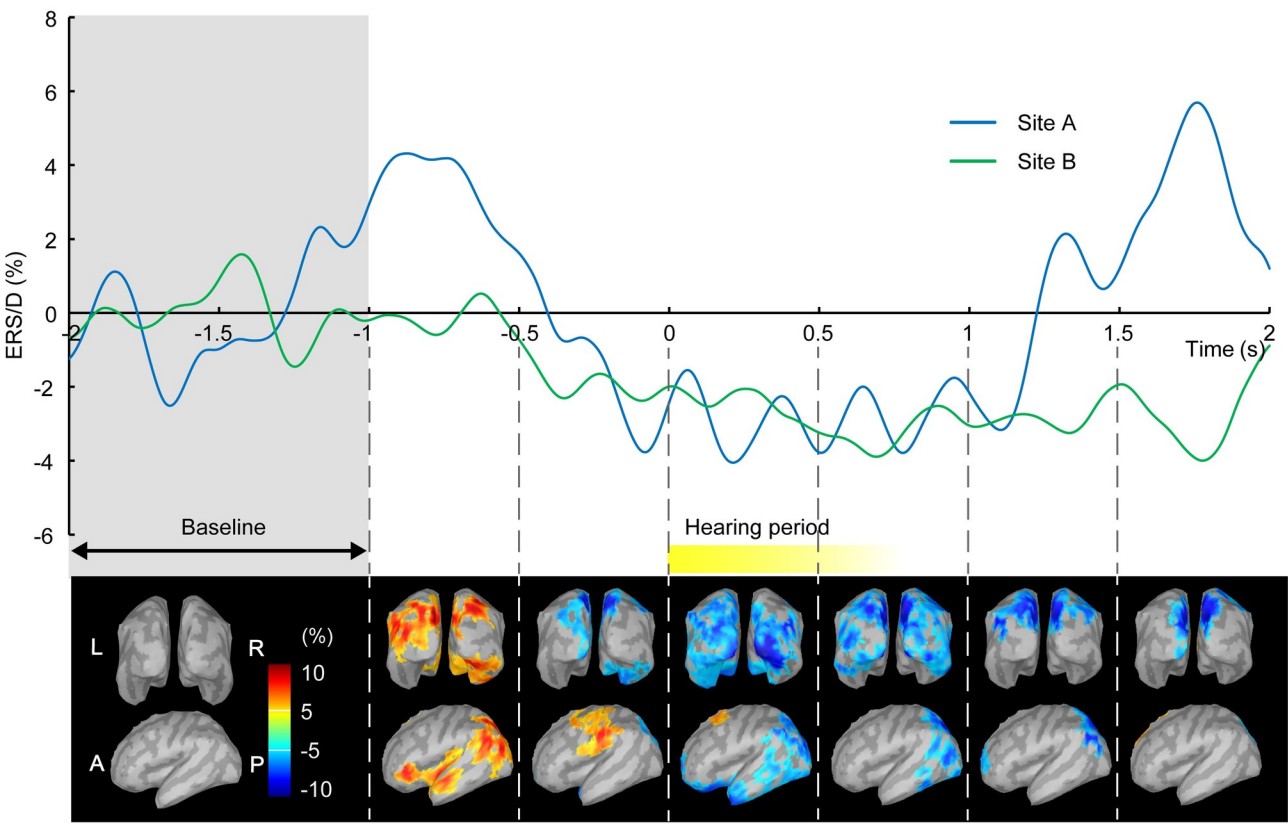

**Fig 3. Amplitude modulation of alpha-band rhythm during face-to-face conversation.** redThe brain responses when two subjects faced each other via the A/V devices and spoke words in turns are shown. Mean alpha rhythm amplitude across 128 speech exchanges was normalized by the mean amplitude over the baseline period, from -2 and -1 s, to calculate Event Related Synchronization (ERS) and Event Related Desynchronization (ERD). The time traces of ERS/D averaged over the whole brain of each subject at site A (blue line) and B (green line) are shown on the upper panel. The brain activity of the subjects at both sites reflects that which is associated with listening, with time point 0 ms being the moment of speech onset of the opposite party. ERD of alpha rhythm is exhibited just before and during hearing the speech (mean: 0.7 s) of the opposite party. The brain surface images on the lower part show mean distributions of ERS (red; <+5%) and ERD (blue; >-5%) on each of the 15,002 vertices across both subjects; back-view (upper row) and left side-view (lower row). This mean alpha rhythm ERS/D was furthermore averaged temporally within each 0.5 s bin. A distinct ERD in the bilateral occipital region (visual area) and left temporal region (linguistic area) observed after 0 s indicate functional involvement of both the visual and auditory systems, suggesting that each subject could visually predict the onset of the opposite party's speech. Abbreviations. L: Left, R: Right, A: Anterior, P: Posterior.

via ADL is another achievement that has hitherto not been reported. The only other existing MEG hyperscanning system that might have comparable video delay is one reported by Hirata et al. [35]. That system comprises two MEGs co-located in one shielded room, with one MEG designed for adults, and the other designed for infants or small children, thus permitting parent-child hyperscanning. The co-location of the MEGs in the same room allows the audio communication to be transmitted directly through the air. However, the two MEGs are designed for recording subjects in supine positions, and thus facial communication with their system has been accomplished similarly to us with video signals transmitted via cameras and projectors. Correspondingly, although the exact amount has not been reported, the co-located MEG hyperscanning system reported by Hirata et al. must certainly have delay in the video signals. Furthermore, the co-location of the subjects in the same room and their auditory communication through air not only means that auditory signals likely precede video signals, but also that the audio cannot be isolated and properly synchronized to the video signals. Finally,

their system is limited in that hyperscanning can only be performed between an adult and a child. Our MEG hyperscanning system realizes real-time video and audio communication between two adults, and uses a more natural, face-to-face, seated orientation (Fig 1). Combined with the extremely low video latency and audio-video synchronization, our system should permit natural conversation. See the S1 Appendix for information about ways that latency and jitter could be reduced even further.

As MEG is silent, and completely non-invasive, our system should permit cortical-level investigation into numerous kinds of subtle and dynamic brain processes which occur during natural two-way communication. For example, our system could be used to measure cortical brain response associated with changes in speech patterns and facial expressions between the participating subjects. The ability to measure this is important as brain responses during dynamic real-time conversation may be quite different than isolated event-related responses. Indeed, consider that the N400 event-related potential component associated with semantic processing of a single word is generally observed about 400 ms after the word is presented [36]. In contrast, responses in everyday conversation have been reported to occur in as little as 200 ms after a conversation partner's speech onset [37]. In addition, a prominent response in the occipital cortex to another's blink has been observed at 250 ms, and this brain response is positively correlated with empathic concern in the viewer [38, 39]. These kinds of fast brain responses that occur back and forth in real-time communication likely have neurocorrelates in both the sender and the receiver, and thus require high temporal resolution hyperscanning to adequately capture. Moreover, it is important to recognize that in natural, two-way communication, both parties alternate between being the sender and the receiver of auditory and visual information, and the brain regions involved when sending (inferior parietal lobule/sulcus, ventral premotor cortex) and receiving (ventral medial prefrontal cortex) communication are different [40]. Therefore, high spatial resolution is also very important in a hyperscanning system, thereby making MEG a preferable modality for investigating the neurocorrelates of natural communication.

Finally, we would like to highlight the importance of the intermediate devices used to transmit/receive audiovisual signals in hyperscanning systems. The quality of these devices and the validation of their signal processing latencies and characteristics is essential for realizing well-controlled experimental designs in neuropsychophysiological experimentation. Moreover, the minimization of the latency through these intermediate devices, such as via a direct fiber optic connection, is a fundamental priority for hyperscanning research protocols in any modality, not only MEG.

Comprehensively, the establishment and verification of our new MEG hyperscanning system opens the door to a new line of neuroimaging research regarding human communication. Future studies employing our system may shed light on the pathophysiology of neurological and psychiatric disorders that manifest with communication deficits, and inspire development of novel medications or interventions.

## Supporting information

**S1 Fig. TTL latency data (photo).** TTL measured by a digital spectrum analyzer. Top: loop-back value, bottom: direct measurement value. doi:10.6084/m9.figshare.19127282.
(TIF)

**S2 Fig. Auditory latency data (photo).** Auditory latency measured by a digital spectrum analyzer. Top: with delay value, bottom: direct measurement value. doi:10.6084/m9.figshare.14872785.
(ZIP)

**S1 File. Video latency data.** Mat files of video latency measured by MEG Acq of the MEG hyperscanning system at Hokkaido University. doi:10.6084/m9.figshare.14872827. (ZIP)

**S2 File. Electrophysiological data.** Fiff files of electrophysiological data measured by MEG Acq of the MEG hyperscanning system at Hokkaido University. doi:10.6084/m9.figshare. 19127285. (ZIP)

**S1 Appendix. Measurement of visual event related field.** A proposal for measuring visual evoked field with high accuracy based on jitter and latency of visual signals transmission. (PDF)

## Author Contributions

**Conceptualization:** Atsushi Shimojo, Kazuyori Yagyu, Koichi Yokosawa, Takuya Saito.

**Data curation:** Hayato Watanabe, Atsushi Shimojo, Kazuyoshi Takano.

**Formal analysis:** Atsushi Shimojo.

**Funding acquisition:** Kazuyori Yagyu, Koichi Yokosawa, Takuya Saito.

**Investigation:** Hayato Watanabe, Atsushi Shimojo, Kazuyoshi Takano.

**Methodology:** Atsushi Shimojo, Tsuyoshi Sonehara.

**Project administration:** Koichi Yokosawa, Takuya Saito.

**Supervision:** Hideaki Shiraishi.

**Validation:** Kazuyori Yagyu, Koichi Yokosawa, Takuya Saito.

**Visualization:** Hayato Watanabe.

**Writing – original draft:** Hayato Watanabe, Jared Boasen.

**Writing – review & editing:** Atsushi Shimojo, Kazuyori Yagyu, Tsuyoshi Sonehara, Kazuyoshi Takano, Jared Boasen, Hideaki Shiraishi, Koichi Yokosawa, Takuya Saito.

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
