## [Decision Letter · Decision Letter 0]

4 Jan 2022

PONE-D-21-21236Construction of a fiber-optically connected MEG hyperscanning system for recording brain activity during real-time communicationPLOS ONE

Dear Dr. Yokosawa,

Thank you for submitting your manuscript to PLOS ONE. After careful consideration, we feel that it has merit but does not fully meet PLOS ONE’s publication criteria as it currently stands. Therefore, we invite you to submit a revised version of the manuscript that addresses the points raised during the review process.

We look forward to receiving your revised manuscript.

Kind regards,

Kiyoshi Nakahara, PhD

Academic Editor

PLOS ONE

Journal Requirements:

3. Thank you for stating the following in the Financial Disclosure section: "This Research was supported by Strategic Research Program for Brain Sciences by Japan Agency for Medical Research and Development JP20dm0107567, The Watanabe foundation, and JSPS KAKENHI Grant Number 20H04496. The funders had no role in study design, data collection and analysis, decision to publish, or preparation of the manuscript."

We note that one or more of the authors are employed by a commercial company: "Research and Development Group, Hitachi Ltd." 

4. We note that Figure 1 in your submission contain copyrighted images. All PLOS content is published under the Creative Commons Attribution License (CC BY 4.0), which means that the manuscript, images, and Supporting Information files will be freely available online, and any third party is permitted to access, download, copy, distribute, and use these materials in any way, even commercially, with proper attribution. For more information, see our copyright guidelines: http://journals.plos.org/plosone/s/licenses-and-copyright.

Additional Editor Comments (if provided):

Please follow the comments of reviewers and correct the deficiencies in the paper.

Reviewer 2's comments are particularly important and need to be addressed in order for this paper to be accepted.

Reviewers' comments:

Reviewer's Responses to Questions

**Comments to the Author**

1. Is the manuscript technically sound, and do the data support the conclusions?

Reviewer #1: No

Reviewer #2: Partly

2. Has the statistical analysis been performed appropriately and rigorously? 

Reviewer #1: N/A

Reviewer #2: N/A

3. Have the authors made all data underlying the findings in their manuscript fully available?

Reviewer #1: Yes

Reviewer #2: No

4. Is the manuscript presented in an intelligible fashion and written in standard English?

Reviewer #1: No

Reviewer #2: Yes

5. Review Comments to the Author

Reviewer #1: In the present paper, the authors reported the details of the Hyperscanning MEG system. Information on how to build up the system, especially information about what devices are used in the integrated system, is usually unclear in the most of hyperscanning systems. Therefore, the present paper would be informative for potential readers who would like to prepare the hyperscanning MEG. However, unfortunately, I have several concerns with this paper.

First and the most important thing is that the scientific significance of this paper is unclear. A low latency to transmit signals is very important. I agree. But when the two devices are apart from each other, and connected via some transmitters, the lag cannot be zero. The hyperscanning EEG and NIRS are better in this respect, because it is possible to record two brains using one device, and two participants could do communication directly without any audio/video devices. Potential readers want to know how much better the author’s hyperscanning MEG compared to old-type hyperscanning MEG especially in ability to depict the inter-brain synchrony. The comparison is indispensable if the author would like to stress the advantage of this setting. What kind of inter-brain effect could be specifically observed only by this hyperscanning MEG? How the subtle inter-brain effect (i.e., inter-brain sync degree) could be specifically captured by the system? In this paper, there is no such results, so this is merely a technical paper: ‘I found new better device, so I integrated it on the recording system. It is better than previous one, because the device is new. I have no idea how the new device contribute to the recording system’. That is the only message I could receive from the paper. For example, the authors could make clear how the small delay, that was in the old-fashioned hyperscanning MEG but not in this present system, affects the detection of inter-brain sync, by doing a small experiment. Because not all hyperscanning MEG system could not be connected via fiber optics, so they are almost always ignoring the delay effect. It would be useful if the authors declare what problems can occurs when there is small latency. Of course, there is nothing wrong with reporting this as a technical report. But I don't think it is a scientific paper.

It is difficult to understand why latency needs to be calculated for reference signals. I understood that the authors want to separate the latency caused by the transmission line (mainly fiber optics) from the latency caused by the microphones and speakers. However, I don't know if my understanding is correct since nothing is mentioned in the paper. Please explain clearly why you need to do this measurement.

The structure of the paper is really far away from standard style: there is no Results section.I think the Results section might be after L204, but please adjust the format. The unconstructed paper is very hard to read.

This is an opinion that has nothing to do with the scientific point of view. Please check the contents of the paper carefully by the authors, before submission. Quality of paper is very low. Here are some examples: sentences starting with L1 have a period at the end for some reason; the content of the paragraph starting with L77 is exactly the same as that starting with L57. I am not a native speaker of English, but I found many points with wrong English grammar. I recommend that the authors send it to an English proofreader before submission.

Minor comments

L228: Latency purple?

L241: What is the ‘mus’?

L243: How did the authors confirm that there is no distortion of sine-wave?

L70 etc.: The term TTL was repeatedly explained.

L85: ‘Photos: with permission by the models.’ It is redundant.

L214: ‘MEG Aqs’ What is the Aqs?

L236: What is the lip-synchronized delay?

L288: The authors said ‘See the Appendix’. Where is your appendix? I could not find it.

Reviewer #2: Watanabe et al. developed a hyperscanning system in two MEG devices with a low latency for audio and video communications. The study presented a newly developed hyperscanning system that can provide the potential to study brain dynamics of social interactions. The systematic configuration was well established and the audio-visual latencies were verified for real-time face-to-face communications. I have minor comments to improve the quality of the manuscript.

- Audio and visual latencies should be measured from the site B to the site A as well for the bidirectional communication. Further, the effect of the direction of the transmission should be investigated.

- An electrophysiological experiment should be performed to use the hyperscanning system in practice. For example, audio, video, and audio-video stimuli are presented at the site A and transmitted to the site B. At the same time, MEG data of a participant are recorded at the site B and the data are analyzed. In this experimental paradigm, electrophysiological signals such as event-related potentials and event-related spectral perturbation for audio, video, and audio-video stimulus can be obtained and the results can verify the system.

6. PLOS authors have the option to publish the peer review history of their article (what does this mean?). If published, this will include your full peer review and any attached files.

Reviewer #1: No

Reviewer #2: **Yes: **Sangtae Ahn

---

## [Author Response · Author response to Decision Letter 0]

8 Mar 2022

See the attachment "Response to Reviewers.docx" file.

---

## [Decision Letter · Decision Letter 1]

4 Apr 2022

PONE-D-21-21236R1Construction of a fiber-optically connected MEG hyperscanning system for recording brain activity during real-time communicationPLOS ONE

Dear Dr. Yokosawa,

Thank you for submitting your manuscript to PLOS ONE. After careful consideration, we feel that it has merit but does not fully meet PLOS ONE’s publication criteria as it currently stands. Therefore, we invite you to submit a revised version of the manuscript that addresses the points raised during the review process.

We look forward to receiving your revised manuscript.

Kind regards,

Kiyoshi Nakahara, PhD

Academic Editor

PLOS ONE

Journal Requirements:

Reviewers' comments:

Reviewer's Responses to Questions

**Comments to the Author**

1. If the authors have adequately addressed your comments raised in a previous round of review and you feel that this manuscript is now acceptable for publication, you may indicate that here to bypass the “Comments to the Author” section, enter your conflict of interest statement in the “Confidential to Editor” section, and submit your "Accept" recommendation.

Reviewer #1: All comments have been addressed

Reviewer #2: All comments have been addressed

2. Is the manuscript technically sound, and do the data support the conclusions?

Reviewer #1: Yes

Reviewer #2: Yes

3. Has the statistical analysis been performed appropriately and rigorously? 

Reviewer #1: Yes

Reviewer #2: Yes

4. Have the authors made all data underlying the findings in their manuscript fully available?

Reviewer #1: (No Response)

Reviewer #2: No

5. Is the manuscript presented in an intelligible fashion and written in standard English?

Reviewer #1: Yes

Reviewer #2: Yes

6. Review Comments to the Author

Reviewer #1: The authors adequately responded to my previous comments. I'm sure that now the authors could get better understanding about the importance of this research project, and about what the authors did to build the hyperscanning MEG system. Now I only have some minor comments.

1.

The authors cited several papers using hyperscanning fMRI system. I recommend that the authors have to select more appropriate papers. First, while the study by Schippers and his colleagues is great, it is not the study of hyperscanning focusing on the interactivity during communication. In this study, two participants could not mutually exchange information. In the case, the video/audio delay does not become a big issue. I suggest that the authors should cite more appropriate literatures investigating neural basis of real social interaction. If the authors think that number of hyperscanning fMRI papers cited in this paper is too much, I strongly suggest some papers by Japanese hyperscanning teams should be replaced by these following papers.

Bilek group

Bilek, E., Ruf, M., Schäfer, A., Akdeniz, C., Calhoun, V. D., Schmahl, C., et al. (2015). Information flow between interacting human brains: identification, validation, and relationship to social expertise. Proc. Natl. Acad. Sci. U.S.A. 112, 5207–5212. doi: 10.1073/pnas.1421831112

Chinese team

https://www.pnas.org/doi/10.1073/pnas.1917407117

Netherland team

https://www.ncbi.nlm.nih.gov/pmc/articles/PMC4280639/

2.

Overall, there is not enough information about what figures show. For example, in Figure 3, the authors showed activation/deactivation (ERS/D) rendered on the surface image. Does the row of surface images in lower panel represent the passage of time? Let us show how the authors calculated the activation on one surface image in detail: which time bin corresponds to the image, how the activation is depicted, whether is this the left hemisphere, and so on. Please also add left-right and AP (anterior-posterior) information. I understand that this small experiment is not the main report of the authors. However, please clearly describe the process that led to these figures and what these figures mean.

Figures in Supporting information could be described a bit more carefully. See, Figure S1. The authors claimed that ‘The loopbacked (round-trip) signal delayed for 7.812 μs (Upper) compared to direct signal (Lower)’, however, I have no idea where these information could be found in the TIFF file.

Reviewer #2: Thanks for the efforts to address my comments. The concerns I had have been fully addressed by the authors.

7. PLOS authors have the option to publish the peer review history of their article (what does this mean?). If published, this will include your full peer review and any attached files.

Reviewer #1: **Yes: **Takahiko Koike

Reviewer #2: **Yes: **Sangtae Ahn

---

## [Author Response · Author response to Decision Letter 1]

18 May 2022

Please see the "ResponseToReviewers" file.

---

## [Decision Letter · Decision Letter 2]

6 Jun 2022

Construction of a fiber-optically connected MEG hyperscanning system for recording brain activity during real-time communication

PONE-D-21-21236R2

Dear Dr. Yokosawa,

We’re pleased to inform you that your manuscript has been judged scientifically suitable for publication and will be formally accepted for publication once it meets all outstanding technical requirements.

Kind regards,

Kiyoshi Nakahara, PhD

Academic Editor

PLOS ONE

Additional Editor Comments:

Several minor typographical errors were noted by the reviewer. Please check the entire manuscript upon receipt of the galley proof and make any final typographical corrections.

Reviewers' comments:

Reviewer's Responses to Questions

**Comments to the Author**

1. If the authors have adequately addressed your comments raised in a previous round of review and you feel that this manuscript is now acceptable for publication, you may indicate that here to bypass the “Comments to the Author” section, enter your conflict of interest statement in the “Confidential to Editor” section, and submit your "Accept" recommendation.

Reviewer #1: All comments have been addressed

2. Is the manuscript technically sound, and do the data support the conclusions?

Reviewer #1: Yes

3. Has the statistical analysis been performed appropriately and rigorously? 

Reviewer #1: N/A

4. Have the authors made all data underlying the findings in their manuscript fully available?

Reviewer #1: Yes

5. Is the manuscript presented in an intelligible fashion and written in standard English?

Reviewer #1: No

6. Review Comments to the Author

Reviewer #1: The authors have adequately replied to my comments, and I think the paper is ready to publish.

The authors have to carefully check there are no any typos.

Here I list typos I found.

L2: Real-time. face-to-face -> Real time face-to-face

L69: The transistor-transistor logic (TTL) -> It seems redundant, because the abbreviation was shown in the manuscript (L60).

L201: -2000 ms to -1000-ms -> -2,000 ms to -1,000 ms

7. PLOS authors have the option to publish the peer review history of their article (what does this mean?). If published, this will include your full peer review and any attached files.

Reviewer #1: **Yes: **Takahiko Koike

---

## [Editor Report · Acceptance letter]

10 Jun 2022

PONE-D-21-21236R2 

Construction of a fiber-optically connected MEG hyperscanning system for recording brain activity during real-time communication 

Dear Dr. Yokosawa:

I'm pleased to inform you that your manuscript has been deemed suitable for publication in PLOS ONE. Congratulations! Your manuscript is now with our production department. 

Kind regards, 

on behalf of

Dr. Kiyoshi Nakahara 

Academic Editor

PLOS ONE